# Molecular diagnosis of scabies using a novel probe-based polymerase chain reaction assay targeting high-copy number repetitive sequences in the *Sarcoptes scabiei* genome

**Lena Chng**[1], **Deborah C. Holt**[2,3], **Matt Field**[4,5], **Joshua R. Francis**[2,6], **Dev Tilakaratne**[6,7], **Milou H. Dekkers**[8], **Greg Robinson**[1], **Kate Mounsey**[9], **Rebecca Pavlos**[10], **Asha C. Bowen**[2,10,11], **Katja Fischer**[1], **Anthony T. Papenfuss**[12], **Robin B. Gasser**[13], **Pasi K. Korhonen**[13], **Bart J. Currie**[2,6], **James S. McCarthy**[1], **Cielo Pasay**[1] *

1 QIMR Berghofer Medical Research Institute, Brisbane, Australia, 2 Menzies School of Health Research, Charles Darwin University, Darwin, Australia, 3 College of Health and Human Sciences, Charles Darwin University, Darwin, Australia, 4 Centre for Tropical Bioinformatics and Molecular Biology and Australian Institute of Tropical Health and Medicine, James Cook University, Cairns, Australia, 5 Genome Informatics, John Curtin School of Medical Research, Australian National University, Canberra, Australia, 6 Royal Darwin Hospital, Tiwi, Australia, 7 Darwin Dermatology, Tiwi, Australia, 8 Queensland Animal Science Precinct, University of Queensland, Gatton, Australia, 9 University of Sunshine Coast, Sippy Downs, Australia, 10 Wesfarmers Centre for Vaccines and Infectious Diseases, Telethon Kids Institute, University of Western Australia, Perth, Australia, 11 Department of Infectious Diseases, Perth Children's Hospital, Perth, Australia, 12 The Walter and Eliza Hall Institute of Medical Research, Parkville, Australia, 13 Department of Veterinary Biosciences, Melbourne Veterinary School, Faculty of Veterinary Sciences, The University of Melbourne, Parkville, Australia

* Cielo.Pasay@qimrberghofer.edu.au

## Abstract

### Background

The suboptimal sensitivity and specificity of available diagnostic methods for scabies hampers clinical management, trials of new therapies and epidemiologic studies. Additionally, parasitologic diagnosis by microscopic examination of skin scrapings requires sample collection with a sharp scalpel blade, causing discomfort to patients and difficulty in children. Polymerase chain reaction (PCR)-based diagnostic assays, combined with non-invasive sampling methods, represent an attractive approach. In this study, we aimed to develop a real-time probe-based PCR test for scabies, test a non-invasive sampling method and evaluate its diagnostic performance in two clinical settings.

### Methodology/Principal findings

High copy-number repetitive DNA elements were identified in draft *Sarcoptes scabiei* genome sequences and used as assay targets for diagnostic PCR. Two suitable repetitive DNA sequences, a 375 base pair microsatellite (SSR5) and a 606 base pair long tandem repeat (SSR6), were identified. Diagnostic sensitivity and specificity were tested using relevant positive and negative control materials and compared to a published assay targeting

**Data Availability Statement:** All relevant data are within the manuscript and its Supporting Information files.

**Funding:** This study was funded by a National Health and Medical Research Council Grant (Improving Health Outcomes in the Tropical North: A Multidisciplinary Collaboration [HOT NORTH], GNT1131932) [BJC, JMC, ACB]. No author received a salary from this Grant. The funders had no role in study design, data collection and analysis, decision to publish, or preparation of the manuscript.

**Competing interests:** The authors have declared that no competing interests exist.

the mitochondrial *cox1* gene. Both assays were positive at a 1:100 dilution of DNA from a single mite; no amplification was observed in DNA from samples from 19 patients with other skin conditions nor from house dust, sheep or dog mites, head and body lice or from six common skin bacterial and fungal species. Moderate sensitivity of the assays was achieved in a pilot study, detecting 5/7 (71.4% [95% CI: 29.0% - 96.3%]) of clinically diagnosed untreated scabies patients. Greater sensitivity was observed in samples collected by FLOQ swabs compared to skin scrapings.

## Conclusions/Significance

This newly developed qPCR assay, combined with the use of an alternative non-invasive swab sampling technique offers the possibility of enhanced diagnosis of scabies. Further studies will be required to better define the diagnostic performance of these tests.

## Author summary

As scabies control efforts continue to grow, scarcity of diagnostic options hinders success of elimination efforts in endemic areas. Efficiency in large-scale monitoring is further obstructed by invasive sample collection techniques, which are often uncomfortable for patients, and lack sensitivity. We have developed two PCR-based diagnostic assays targeting repetitive DNA elements. These were identified using new data on the *S. scabiei* genome. Targeting these elements by PCR improved the detection of scabies DNA. Enhanced sensitivity was demonstrated when tested against routine microscopy and a published PCR-based diagnostic assay. When combined with a non-invasive, effective FLOQ swab sampling method, the developed qPCR-based assays may provide a useful complementary tool for diagnosis of scabies, and its application will likely improve scabies control in target populations.

## Introduction

Scabies is an ectoparasitic infection of the skin caused by the "itch mite", *Sarcoptes scabiei* [1]. Efforts to control scabies are hampered by limitations of available diagnostic methods, resulting in poor understanding of scabies epidemiology and treatment efficacy [2]. Scabies is routinely diagnosed by clinical assessment, which requires clinical expertise [3,4]. As clinical findings often mimic those of other common skin conditions, such as eczema and bacterial skin infections, clinical diagnosis lacks specificity [5]. Moreover, atypical presentations in certain patients, such as immunocompromised individuals [1], can lead to clinical misclassifications and missed detections. Apart from clinical assessment, diagnosis by microscopy of skin scrapings can be used as a confirmatory parasitologic method [1]. However, microscopy is insensitive due to the low mite numbers in ordinary scabies [6]. Additionally, collection of skin scrapings with a scalpel or sharp needle can be uncomfortable and hence is not well accepted by many patients, especially children. Recent advances in dermatoscopic technologies enable *in-vivo* visualisation of scabies mites, offering promise of increasing diagnostic accuracy [7]. Broad usage of such technology may be restricted by its limited availability, cost and required expertise [8], particularly in areas where burden is highest and health infrastructure limited.

Diagnostic assays based on PCR techniques are increasingly being used to detect viral, bacterial and parasitic infections [9–11], including scabies. Several PCR-based scabies diagnostic assays have been described [12–18]. Conventional PCR techniques require the use of gel electrophoresis for analysis of amplified products, which is labour intensive, particularly when screening large sample numbers [9]. Nested-PCRs can achieve higher sensitivity [15], but this technique is generally avoided due to an increased risk of cross-contamination [19]. Among available diagnostic assays based on PCR, quantitative PCR (qPCR) is increasingly being applied due to its increased sensitivity, rapid cycling times and reduced risk of contamination [11]. Two diagnostic qPCR assays have been described for scabies, both targeting mitochondrial genes, namely the cytochrome c oxidase subunit 1 gene (*cox1)* [13], and the 16S rRNA gene [12]. To date, only the *cox1* assay has been tested in human scabies [13].

New diagnostic PCR assays targeting high copy number repeats have been identified in a number of parasitic helminths by interrogating next generations sequencing (NGS) data [20,21]. Highly repetitive non-coding DNA elements, including microsatellites, have been identified across the *S. scabiei* genome [22], making them an attractive target for diagnostic PCR assays.

The performance of a diagnostic qPCR assay relies on the recovery of the target DNA from the sample, which is affected by both sample collection and extraction methods [23]. For scabies, an alternative, non-traumatic sample collection method, such as swab sampling, can help improve patient acceptability compared to routine skin scraping. Smooth-surfaced swabs have been used in several studies [13,24,25], but problems with sensitivity and sample quantity have been described, for example, with the use of Catch-All swabs [24]. Such problems could be resolved with the use of an alternate swab type, such as flocked swabs e.g. FLOQ, which feature a highly textured surface for a higher sampling yield [26].

The aim of this study was to develop a diagnostic probe-based qPCR assay for scabies targeting novel repetitive DNA elements. First, bioinformatic tools were used to identify novel non-coding repetitive DNA elements as potential targets. Two diagnostic qPCR assays were developed, optimised and compared with a probe-based qPCR assay targeting the scabies mite *cox 1* gene [13]. We evaluated the use of these alternative targets in a porcine scabies model, and compared the performance of non-invasive sampling methods for collection of diagnostic material with two swab types: Catch-All and FLOQ. Finally, we tested the new diagnostic assays in patients from a clinical setting and school children from remote Aboriginal communities in Australia who participated in a school-based surveillance study.

## Methods

### Ethics statement

The porcine scabies model [27] was maintained at Queensland Animal Science Precinct (QASP), The University of Queensland (UQ), Gatton. Maintenance of this model was approved by The University of Queensland Animal Ethics Committees (approval number QIMR/291/18).

The first clinical study conducted in Darwin was approved by the Northern Territory Department of Health, Menzies School of Health and Research (Menzies) Human Research Ethics Committee (HREC) (approval number 2018–3120) and Queensland Institute of Medical Research (QIMR) Berghofer HREC (approval number P2376). Patients, or parent or caregiver, recruited in the study provided written informed consent. The second clinical study conducted in Kimberley, Western Australia [28] was approved by the Child and Adolescent Health Research Ethics Committee (approval number RGS0000000584) and the Western

Australian Aboriginal Health Ethics Committee (approval number 819). Parents or caregivers of children recruited in the study provided written informed consent.

## Study design and population

This study is a prospective study that aimed to develop scabies diagnostic probe-based qPCR assays and test them in patients from two settings, a clinical setting and remote Aboriginal communities in Australia. A schematic of the project workflow is presented in Fig 1.

Patients from the first clinical setting were recruited from Royal Darwin Hospital (RDH) and the Darwin Dermatology (DD) clinic. The inclusion criterion was patients clinically diagnosed with ordinary or crusted scabies. Patients with ordinary scabies satisfied the criteria for confirmed scabies or clinical scabies according to the 2018 International Alliance for the Control of Scabies (IACS) criteria for the diagnosis of scabies (S1 Table) [29]; patients with crusted scabies were diagnosed as per the Northern Territory Department of Health Crusted (Norwegian) Scabies Grading Scale and Treatment Royal Darwin Hospital Plan document [30]. Patients with other scaly skin conditions that did not fit the 2018 IACS criteria for the diagnosis of scabies [29] were used as scabies negative controls for the qPCR assay. Samples were taken from all body sites presenting scabies or alternative skin lesions. In the second study, school children who participated in a school-based surveillance were recruited. The school-surveillance was part of the SToP (See, Treat, Prevent) skin sores and scabies trial conducted in Kimberley, Western Australia. The study design and selection criteria of this study has been described elsewhere [28]. Inclusion criteria were children with suspected scabies according to the 2018 IACS diagnostic criteria for scabies [29].

## Analysis of next generation sequencing data

Repetitive DNA elements were sourced from Illumina reads of *S. scabiei* var. *hominis* (human mite) and *S. scabiei* var. *suis* (pig mite) draft genome sequence reads. Reads were downloaded from Bioproject accession: PRJEB12428 [31] and run through Trim_galore v0.4.5 using

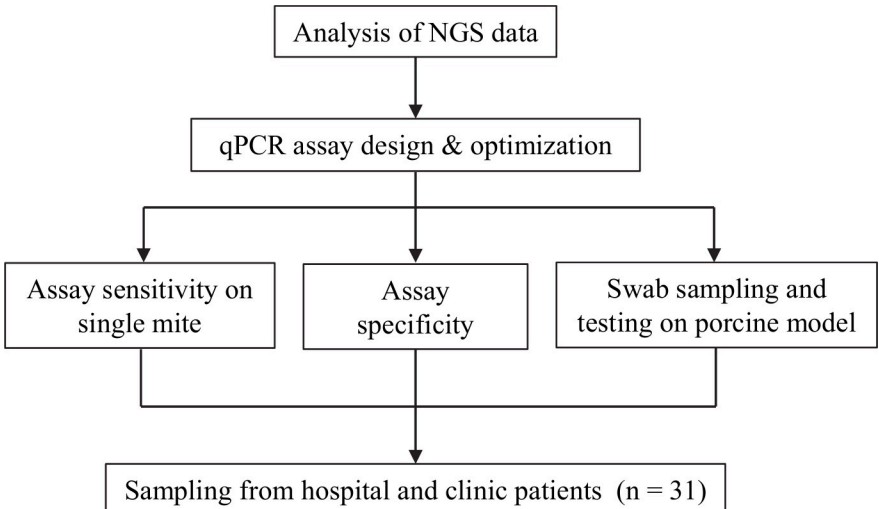

**Fig 1. qPCR assay for scabies diagnosis project flowchart.** qPCR assays were developed from targets sourced from next generation sequencing (NGS) data. Assay optimisation was subsequently carried out through a series of sensitivity and specificity testing. Sampling efficiency of swabbing was then evaluated with the assay using material collected from the pig model. The applicability of the assay was then tested in a clinic/hospital setting, prior to field evaluation.

Cutadapt v1.15 [32] with default parameters for base quality trimming and adapter removal. Human mite samples were aligned to human reference genome GRCh37 while pig mite samples were aligned to pig reference genome susScr2. Reads aligning to their respective hosts were removed using SAMTools v1.1 [33]. Processed reads were submitted to RepeatExplorer2 and repeat analysis was carried out as described by Pilotte *et al.*, 2016 [21]. Highest scoring repeat groups were chosen as potential assay targets. Possible targets were selected from repeats present in both *S. scabiei var. suis* and *S. scabiei* var. *hominis* sequences. This approach was used as the assay was optimised with mites obtained from a porcine scabies model [27] before testing on human samples.

For confirmation, all candidate repeats were aligned to a recent scabies genome assembled from Pacbio reads (NCBI accession identifier WVUK01000000) [34] using BLAST v2.6.0 [35] with all repeats aligning in multiple locations. Raw Pacbio reads were also examined for repeat sequences using RepeatModeler 1.0.11 [36]. All repeats were pooled and input to Multiple Alignment using Fast Fourier Transform (MAFFT) [37] for multiple sequence alignment and a phylogenetic tree construction. To confirm the uniqueness of the RepeatExplorer2 repeats, candidates underwent further analysis using the NCBI Nucleotide BLAST tool under default settings to exclude possible targets in human DNA, common skin bacteria or lice.

**Primer and probe design.** Candidate primer and probe pairings were designed using the Primer Quest online tool https://sg.idtdna.com/Primerquest/Home/Index (Integrated DNA Technologies, Coralville, IA) using the default parameters for qPCR. Probes were labelled with a 6FAM (a carboxyfluorescein fluorescent dye) at the 5' end, a double quencher ZEN in the internal sequence, and 3IABkFQ (an Iowa black quencher) at the 3' end. Cross-reactivity of targets was examined with the Primer-BLAST software (http://blast.ncbi.nlm.nih.gov/Blast.cgi)) [38]. qPCR assay performance was compared to a real-time qPCR assay that targets the *S. scabiei* mitochondrial *cox1* gene (Genbank accession number: KR477839.1) [13]. As the published *cox1* qPCR assay had been designed to target a human scabies mite sequence and our pilot study was undertaken with *S. scabiei var suis*, we designed new primers and probes to accommodate two polymorphisms between pig and human scabies mite sequences for the purpose of this study. Details of all primers used are displayed in the S2 Table.

## Extraction of genomic DNA from mites and clinical samples

Genomic DNA (gDNA) was extracted from live female *S. scabiei* var. *suis* mites isolated from skin crusts obtained from the porcine scabies model. DNA was extracted with the QIAamp DNA mini kit (Qiagen), with some modifications to the manufacturer's protocol. Mites were first homogenised with a motorised micropestle (DWK Life Sciences, NJ, USA) for 2mins. Homogenised samples were then incubated at 37˚C for 30 minutes with 200 μL ATL buffer; then, 200 μL AL buffer and 20 μL Proteinase K were added to the samples and incubated overnight at 56˚C. Subsequent steps were carried out according to the kit manual.

Mite DNA from the FLOQ swabs was extracted with the QIAamp DNA mini kit (Qiagen), with some modifications to the manufacturer's protocol. Samples were first spiked with 3 μL qPCR extraction control Red (Bioline), and then incubated at 37˚C for 30 minutes and vortexed for 3 minutes to dislodge mite material from swab. Swabs were removed and 300 μL AL buffer and 20 μL Proteinase K were added. All samples were incubated overnight at 56˚C. Subsequent steps were carried out according to the kit's manual. Extraction of mite DNA from skin scrapings involved an additional homogenisation step after the initial 37˚C incubation with ATL buffer. Homogenisation was carried out with the Precellys 24 homogeniser at 6800 rpm for 20 seconds. Extraction Control Red (Bioline) was added to each reaction for detection of the extraction control.

**Cloning of target sequences into plasmid vector.** Target sequences were PCR amplified, cloned and sequenced from gDNA extracted from live female *S. scabiei* var. *suis* mites. PCR reactions included 1×HotStarTaq MasterMix (Qiagen, Hilden, Germany), 0.2 μm of the specified forward and reverse primers, PCR-grade water and 3.5 μL DNA template. Thermocycling was performed with Veriti 96-well Thermal Cycler (Applied Biosystems, CA, USA). Cycling conditions were 95˚C for 15 minutes, then 35 cycles of 94˚C for 45 seconds, 50˚C for 45 seconds and 72˚C for 1 minute; and a final extension of 72˚C for 10 minutes. Amplified products were visualised with a 1.5% agarose gel. PCR products were gel-purified, cloned into the pGEM-T Easy Vector system (Promega Corporation, WI, USA) and sequenced using the Big-Dye Terminator v3.1 Cycle Sequencing Kit (Applied Biosystems). DNA concentration of linearised recombinant plasmid was determined using the Qubit 1× dsDNA HS Assay Kit (Invitrogen, CA, USA).

**qPCR assay.** Optimal assay conditions for the diagnostic qPCR assays were determined to be an initial 2 minutes denaturation for 95˚C, followed by 35 cycles of 95˚C for 5 seconds and 50˚C for 5 seconds using an optiCFX384 Touch QPCR Detection System. Each qPCR reaction (10 μL) included 1× QuantiNova Probe PCR Mix (Qiagen), 0.4 μm forward and reverse primers, 0.2 μm probe and nuclease-free water. Reaction efficiency (E) was assessed by generating a standard curve, which plotted the cycle quantification (Cq) value against the $\log_{10}$ of the linearised plasmid DNA template concentration at each dilution. The coefficient of determination ($R^2$) and slope were then determined from the standard curve using the software from the qPCR apparatus (CFX manager v 3.1, Bio-Rad Laboratories, CA, U.S.A). Two dilution series were prepared by serial dilutions of linearised plasmid DNA. The first series consisted of 10-fold dilutions from $3\times 10^8$ to $3\times 10^1$ copies/μL and the second consisted of two-fold dilutions from $3\times 10^1$ to 0.9375 copies/μL. The limit of detection (LOD) was determined by testing all dilution series in triplicate. qPCR assay LOD was derived as the last dilution where all replicates are positive.

**Assay sensitivity.** Genomic DNA (gDNA) extracted from a single mite was serially diluted 10-fold from $10^{-1}$ to $10^{-3}$. All dilutions were also tested using *cox1* qPCR and the performance of the new assays compared to this reference assay. All samples were tested in triplicate under the qPCR conditions described above. Absolute DNA copy numbers were determined from Cq values using the standard curves generated to determine the LOD.

**Assay specificity.** The specificity of the two qPCR assays was tested by both conventional end-point PCR assay and by qPCR. Genomic DNA extracted from 100 house dust mites (HDMs), *Dermatophagoides farina*, was serially diluted 10-fold to replicate ten HDMs and one HDM. Other species tested include *Pediculus humanus* (human head lice), *Demodex canis* (related to the human eyelash mite), *Psoroptes ovis* (sheep scab mite), and common skin bacterial (*Staphylococcus aureus*, *Streptococcus pyogenes* and *Propionibacterium acnes)* and fungal (*Trichophyton sp.*—unidentified species, *T. rubrum* and *T. interdigitale)* pathogens. Uninfected pig skin and human skin were also tested for specificity of the PCR reaction.

## Testing of swab sampling methods in the scabies porcine model

The sampling efficiency of non-invasive sample collection methods for scabies was evaluated on a porcine scabies model. FLOQ (Copan Diagnostics, CA, USA) and Catch-All (Epicentre Biotechnologies, WI, USA) swab samples were collected from scabies infected pigs from the porcine scabies model. Four pigs were sampled on both ears, at three locations per ear, and a total of 24 FLOQ swab samples and 24 Catch-All swab samples were collected. Uninfected pig skin was used as negative control. Saline-moistened swabs were rubbed on peripheral areas of both ears in three different locations (top, middle and bottom). Swabs were submerged in tubes containing 300 μL ATL buffer.

## Collection and analysis of clinical samples

For the first study, samples were collected from patients recruited at two clinical sites in Darwin: Royal Darwin Hospital and Darwin Dermatology. Two types of samples were collected from each individual: skin scrapings and FLOQ swabs. Skin scrapings were subjected to both microscopy and qPCR, while FLOQ swabs were only subjected to qPCR. Scrapings were collected from skin lesions using a scalpel blade, and placed into plastic containers and sent to Menzies School of Health Research (MSHR). After microscopy, the samples were then transferred into 1.5 mL tubes with 300 μL ATL buffer for DNA extraction and qPCR testing. Swab samples were collected by rubbing saline-moistened swabs on the surface of skin lesions, which were then transferred to a 1.5 mL tube containing 300 μL ATL buffer for DNA extraction and qPCR. All samples were stored at -20˚C between collection and DNA extraction.

Microscopic examination of the skin scrapings was carried out at MSHR by trained research technicians prior to addition into ATL buffer. The skin scrapings were observed directly under a dissecting microscope, and a positive microscopy diagnosis was defined by the observations of scabies mites or eggs. Samples collected from patients with non- scabies skin conditions (no scabies group) were not subjected to microscopy.

DNA extracted from clinical samples were tested in triplicate using the two new assays and the *cox 1* assay. A positive result was only assigned to a sample when two of the three replicates had a Cq value that crossed the pre-determined threshold. For patients with more than one sample taken from scabies lesions in different body sites, a positive diagnosis was allocated if one of the samples were positive by qPCR.

For the second study, undertaken in remote Indigenous communities, FLOQ swabs samples were collected by rubbing saline-moistened swabs on the surface of skin lesions, which were then transferred to a 1.5 mL tube containing 300 μL ATL buffer stored at 4˚C. After sample collection in the field, all swab samples were stored at -20˚C prior to shipment to QIMR, where DNA extraction and qPCR testing were performed.

## Statistical analysis

Paired t-tests were used to test for differences in qPCR Cq values when appropriate, and p values of less than 0.05 were considered significant. All data analysis was carried out on the Graphpad Prism 7.0e software. Assay sensitivity and specificity were estimated through the cross tabulation of test results [39]. Exact Clopper-Pearson 95% confidence intervals (CI) for sensitivity and specificity values were calculated with the MedCalc (v19.1) software.

## Results

### Identification of repetitive sequences

RepeatExplorer2 analysis of Illumina reads from the two human mite libraries yielded seven and nine candidate repeats, while 13 and 15 candidate repeats were generated from the two pig mite libraries. All repeats identified were classified as microsatellites with the exception of a single long tandem repeat (LTR) found in all four libraries. Phylogenetic analysis of the repeats revealed six distinct clusters of repeats found in all four libraries and two repeats specific to the pig mite libraries. All low-confidence scoring repeat groups were next filtered out resulting in the removal of some candidates. No repeats identified by RepeatModeler were taken forward due to lack of overlap with RepeatExplorer2 Illumina based repeats. The eleven high-confidence repeats present in both the *S. scabiei var. suis* and *S. scabiei var. hominis* genomes, comprising the single LTR and 10 microsatellites (S3 Table) were verified by

alignment to a scabies genome assembled from PacBio-generated reads [34]. Following filtering with the NCBI Nucleotide BLAST at default parameters, two clusters with sequence lengths that permitted design of primer pairs and probes for an optimal amplicon size of < 150bp were selected for qPCR assay development. These comprised a 375 bp microsatellite (SSR5) and a 606bp LTR (SSR6) (S3 Table), and were selected as candidate targets for qPCR assays; primers and probes were then designed to selectively amplify these targets (S2 Table).

**Assay sensitivity.**   The LOD of the two qPCR assays was first determined by testing two dilution series of linearised plasmid DNA stocks of each candidate assay target, SSR5 and SSR6. Both assays covered a range of eight 10-fold dilutions from $3 \times 10^8$ copies/µL to $3 \times 10^1$ copies/µL. The lower LOD was then further determined by testing serial two-fold dilutions from $3 \times 10^1$ copies/µL to 0.9375 copies/µL. The LOD (defined as the last dilution where all replicates were positive) was determined as 3.8 copies per 10 µL reaction for the SSR5 assay and 1.9 copies per 10 µL reaction for the SSR6 assay (S1 Fig). Individual Cq values of each replicate tested in the two-fold dilution series are presented in the S4 Table. Using CFX manager, the reaction efficiency for each assay was calculated at 95.8% for the SSR5 assay and 96.0% for the SSR6 assay. Assay sensitivity was then assessed against a biologic target using serially 10-fold diluted genomic DNA extracted from a single *S. scabiei* var. *suis* mite. DNA was detected up to a 1:100 dilution (0.028 ng/µL), with a mean Cq value of 31.54± 0.66 for the SSR5 assay and 32.42± 0.09 for the SSR6 assay. The mean Cq value at 1:100 dilution for the *cox1* assay fell between the two new assays at 32.08± 0.28 (Fig 2). Individual Cq values of triplicate reactions are presented in the S5 Table.

**Assay specificity.**   Assay specificity was evaluated against serial 10-fold diluted genomic DNA of house dust mites (*D. pteronyssinus*), human body louse (*P. humanus*), the dog follicle mite (*D. canis*), the sheep scab mite (*P. ovis)* (8.32 ng/µL), and the following bacterial and skin pathogens of humans: *S. pyogenes*, *S. aureus*, *P. acnes*, *Trichophyton sp.*, *T. rubrum*, *T. interdigitale*, and host DNA (human and pig skin). No amplification occurred in any of these samples (S2 Fig).

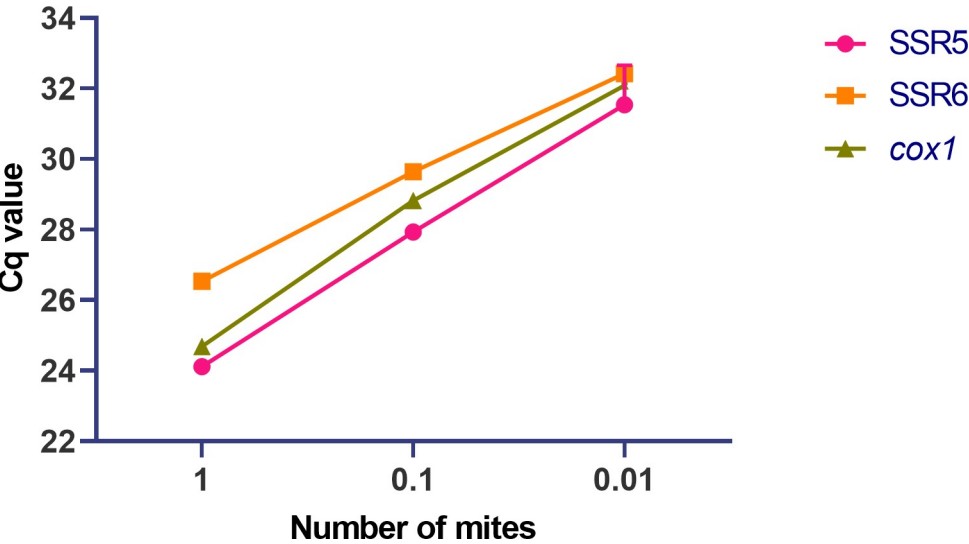

**Fig 2. Sensitivity of probe-based qPCR assay detection on mite gDNA targeting SSR5, SSR6 and *cox1*.** Mean cycle quantification (Cq) values reported for the amplification of serial single mite dilutions with the SSR5 and SSR6 assays and the reference assay *cox1*. Each point represents the mean of triplicate reactions. No amplification was detected in the 1:1000 dilution of the genomic DNA extracted from a single mite.

## Comparison of assay performance on samples collected using FLOQ and Catch-All swabs

All 24 samples collected from the porcine scabies model using both the FLOQ and 24 Catch-All swabs were qPCR-positive in both the SSR5 and the SSR6 assays. Lower Cq values were obtained on samples collected using FLOQ swabs compared to those collected using Catch-All swabs, suggesting superior sampling. In the SSR5 assay, FLOQ swabs reported 7351 DNA copies per sample, whereas Catch-All swabs reported 3782 DNA copies per sample (p = 0.004). In the SSR6 assay, FLOQ swabs reported 2577 DNA copies per sample, whereas Catch-All reported 1272 DNA copies per sample (p = 0.004) (Fig 3). No amplification was observed in the negative controls.

## Testing of patient samples

In total, 31 patients were enrolled between January and August 2019 at RDH (n = 5 patients) and DD (n = 26 patients) (Fig 4). Of these, 8 (25.81%) were of Australian Indigenous ethnicity and the remaining 23 (74.19%) were non-Indigenous. The median age was 55.5 years (interquartile range, IQR: 29–67). Each patient had two samples taken: skin scraping (n = 31 samples) and FLOQ swab (n = 31 samples). Additional samples were taken from three patients at RDH due to the presence of scabies lesions on a second body site (n = 3 skin scrapings samples and n = 3 FLOQ swabs samples). Thus, a total of 68 samples were collected from the 31 patients. From the 31 patients included in the study, 12 were diagnosed with clinical scabies at the time of sample collection, and 19 patients were diagnosed with other skin conditions, including psoriasis (n = 7), dermatitis (n = 4), eczema (n = 2), seborrhoeic keratoses (n = 2), Bowen's disease (n = 1), tinea nigra (n = 2), and pityriasis versicolor (n = 1). These 19 subjects constituted the 'no scabies' clinical group. The 12 patients diagnosed with clinical scabies were further divided

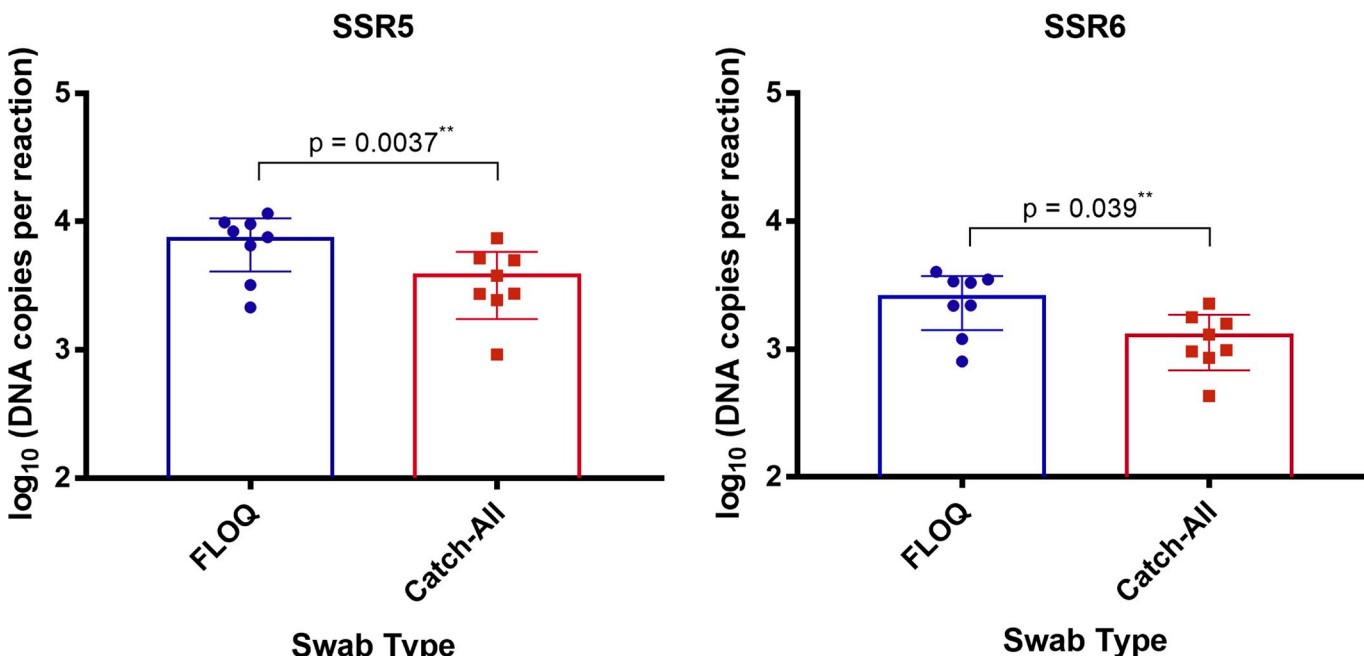

**Fig 3. Comparative testing of FLOQ and Catch-All swabs.** Calculated DNA quantity (DNA copies per reaction) in samples collected by FLOQ and Catch-All swabs using the **(A)** SSR5 and **(B)** SSR6 assays. Standard curves constructed from assays dilution series were used to determine quantity of genomic DNA, represented as DNA copies per reaction. Each data point represents DNA recovery from one pig ear. The calculated quantity of target DNA recovered using FLOQ swabs was higher than using the Catch-All swabs in both assays. (All DNA copies reported are the mean values of triplicate reaction tested).

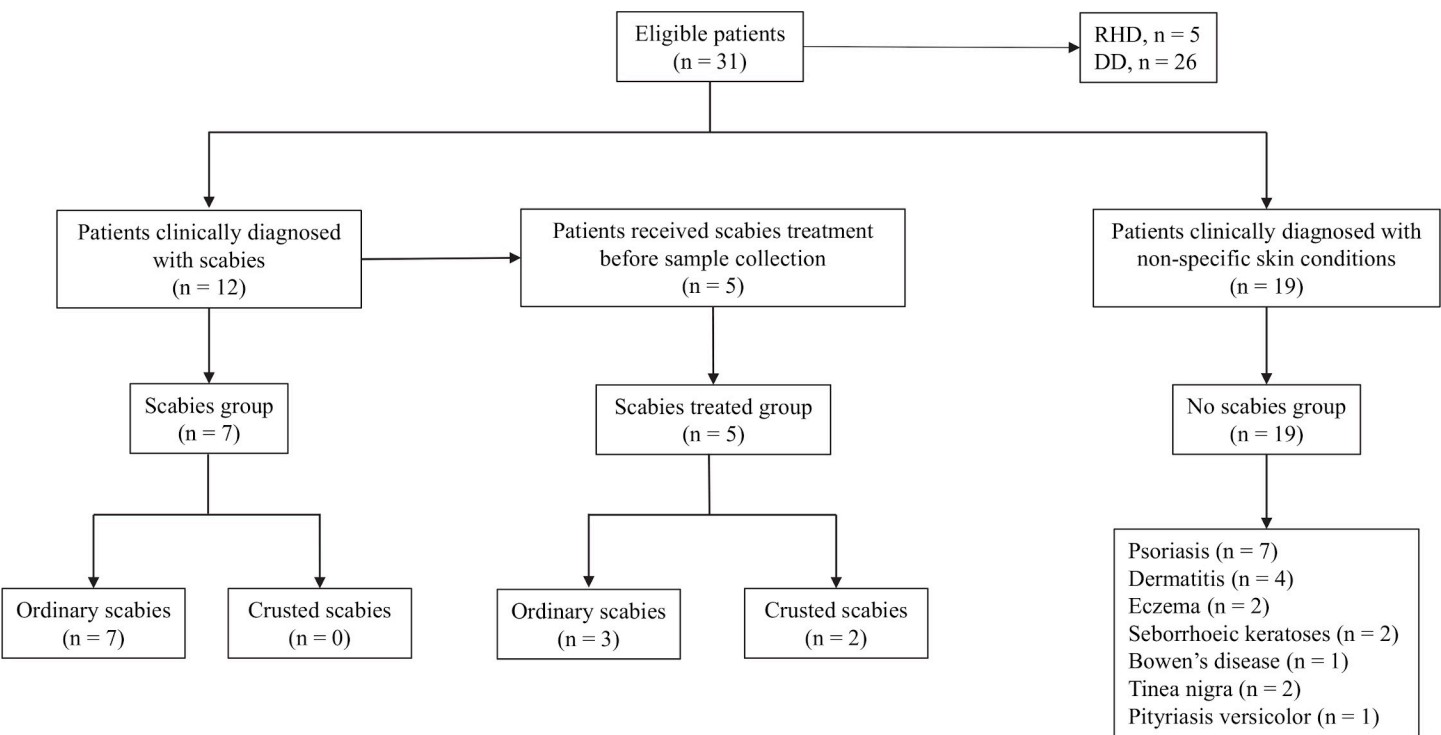

**Fig 4. Flowchart classification of patients from RDH and DD.** Patients were distributed in three groups: scabies group, which included patients clinically diagnosed with scabies who had not received treatment at the time of sampling; treated scabies group, which included patients clinically diagnosed with scabies who had received treatment before sampling; no scabies group, which included patients clinically diagnosed with non-specific conditions. RHD: Royal Darwin Hospital; DD: Darwin Dermatology clinic.

into two groups, one having received treatment prior to sample collection (n = 5) and the other not having received any treatment prior to sample collection (n = 7) (Table 1).

## Assay performance on clinical samples

For a qPCR diagnosis of scabies, two of the three qPCR assays (SSR5, SSR6, *cox1)* were required to be positive. Among the seven patients with a clinical diagnosis of scabies who had not received treatment, five of the seven samples collected by the FLOQ swab tested positive, resulting in a 71.4% sensitivity (95% CI: 29.0% - 96.3%); all three qPCR assays (SSR5, SSR6 and *cox1)* were positive in these five samples (Fig 5). Among this group, skin scrapings of only two of seven were positive by microscopy. qPCR undertaken on the skin scrapings showed positive results in 3/7 subjects (sensitivity of 42.9% [95% CI: 9.9% - 81.6%]). Two of the five patients with previous treatment for scabies tested positive on the FLOQ swab samples, again in all three assays (sensitivity 40% [95% CI: 5.3% - 85.3%]); one of five was positive by microscopic diagnosis on skin scrapings, while two of the five patients in this group had a positive qPCR result (Fig 6). Of note, a total of five qPCR positive patients (4 untreated and 1 treated), were not detected by microscopy. All samples collected from the 19 patients suffering from non-scabies skin conditions, tested negative in the SSR5, SSR6 and *cox1* assays, resulting in a specificity of 100% (95% CI: 82.4% - 100%).

To explore the relative performance of the three qPCR assays (SSR5, SSR6 and *cox1*) the Cq values of all positive samples were compared (Table 2). In agreement with the pattern observed in mite genomic DNA testing and pig samples, the relative order of sensitivity by Cq value was SSR5> *cox1>* SSR6 (mean differences in Cq values of SSR5: cox 1 of 1.03 ± 1.73 and cox1:

**Table 1. Demographics and disease history of patients from RDH and DD enrolled in this study.**

| Patient No. | Age | Indigenous | Clinical Diagnosis | Duration of Symptoms | Scabies Treatment | Course of Treatment | Collection Site | Microscopy Diagnosis | Sample Source |
|---|---|---|---|---|---|---|---|---|---|
| 1 | 69 | Yes | Ordinary Scabies* | 3 months | Permethrin & Ivermectin | 3 months | Right Upper Leg | Negative | DD |
| 2**** | 4 moths | Not Stated | Ordinary Scabies | 1 week | No | N.A. | Left Foot | Negative | RDH |
| | | | Ordinary Scabies | 1 week | | N.A. | Right Foot | Positive | RDH |
| 3**** | 16 | Yes | Ordinary Scabies | > 1 month | No | N.A. | Right Wrist | Negative | RDH |
| | | | Ordinary Scabies | > 1 month | | N.A. | Left Axilla | Positive | RDH |
| 4 | 2 months | Yes | Ordinary Scabies | 2 weeks | Permethrin | 4 days | Left Foot | Positive | RDH |
| 5 | 78 | No | Ordinary Scabies | 2 months | No | N.A. | Right Palm | Negative | DD |
| 6 | 11 | No | Ordinary Scabies | 4 weeks | No | N.A. | Right Thumb | Negative | DD |
| 7 | 60 | No | Ordinary Scabies | 3 months | No | N.A. | Left Palm | Negative | DD |
| 8 | 9 | No | Ordinary Scabies | 3 weeks | No | N.A. | Left Finger | Negative | DD |
| 9**** | 2mos | Yes | Ordinary Scabies | > 2 weeks | Permethrin | > 2 Weeks | Abdomen | Negative | RDH |
| | | Yes | Ordinary Scabies | > 2 weeks | | | Left Foot | Negative | RDH |
| 10 | 55 | Yes | Grade 1 Crusted Scabies** | Several months | Permethrin | 2 weeks | Left Palm | Negative | DD |
| 11 | 65 | No | Grade 2 Crusted Scabies*** | Several weeks | Benzyl Benzoate & Ivermectin | 2 days | Right Elbow | Positive | RDH |
| 12 | 91 | Yes | Ordinary Scabies | 3 months | No | N.A. | Right Arm | Negative | DD |
| 13 | 68 | No | Dermatitis | 6 months | N.A. | N.A. | Left Palm | N.A. | DD |
| 14 | 39 | No | Dermatitis | 3 months | N.A. | N.A. | Left Knee | N.A. | DD |
| 15 | 78 | No | Bowen's Disease | 6 months | N.A. | N.A. | Left Calf | N.A. | DD |
| 16 | 1mo | No | Eczema | 12 months | N.A. | N.A. | Right Palm | N.A. | DD |
| 17 | 55 | No | Seborrhoeic keratoses | Years | N.A. | N.A. | Right Arm | N.A. | DD |
| 18 | 32 | No | Tinea nigra | 1 year | N.A. | N.A. | Right Palm | N.A. | DD |
| 19 | 39 | No | Seborrhoeic dermatitis | 3 months | N.A. | N.A. | Midline Scalp | N.A. | DD |
| 20 | 46 | No | Dermatitis | 12 months | N.A. | N.A. | Anterior Neck | N.A. | DD |
| 21 | 38 | Yes | Pityriasis versicolor | Several years | N.A. | N.A. | Right Flank | N.A. | DD |
| 22 | 66 | No | Psoriasis | NA | N.A. | N.A. | Dorsum of hand | N.A. | DD |
| 23 | 55 | No | Psoriasis | NA | N.A. | N.A. | Left Thigh | N.A. | DD |
| 24 | 59 | No | Psoriasis | NA | N.A. | N.A. | Left Leg | N.A. | DD |
| 25 | 56 | No | Psoriasis | NA | N.A. | N.A. | Right Elbow | N.A. | DD |
| 26 | 84 | No | Dermatitis | NA | N.A. | N.A. | Mid Back | N.A. | DD |
| 27 | Not Stated | No | Plaque Psoriasis | NA | N.A. | N.A. | Left Forearm | N.A. | DD |
| 28 | 67 | No | Psoriasis | NA | N.A. | N.A. | Right Knee | N.A. | DD |
| 29 | 80 | No | Psoriasis | NA | N.A. | N.A. | Right Arm | N.A. | DD |
| 30 | 33 | No | Eczema | NA | N.A. | N.A. | Left Elbow | N.A. | DD |
| 31 | 29 | No | Tinea | NA | N.A. | N.A. | Right Forearm | N.A. | DD |

Patients were distributed in three groups: scabies group, which included patients clinically diagnosed with scabies who did not receive treatment at the time the samples were taken; scabies group treated, which included patients clinically diagnosed with scabies who received treatment before samples were taken; no scabies group, which included patients clinically diagnosed with non-specific conditions.

*Ordinary Scabies: infestation with *Sarcoptes scabiei* var. *hominis*, in the absence of features of crusted scabies (hyperkeratotic plaques or scaling with abundant mites) (Engelman *et al*., 2018)

**Grade 1 crusted scabies: infestation with *Sarcoptes scabiei* var. *hominis* with hyperkeratotic plaques or scaling with abundant mites, total score of 4–6

***Grade 2 crusted scabies: infestation with *Sarcoptes scabiei* var. *hominis* with hyperkeratotic plaques or scaling with abundant mites, with total score of 7–9 (Davis *et al*., 2013)

****Patient 2, 3 and 9 had an additional sample taken due to the presence of a second scabies lesion.

DD: Darwin Dermatology; RDH: Royal Darwin Hospital

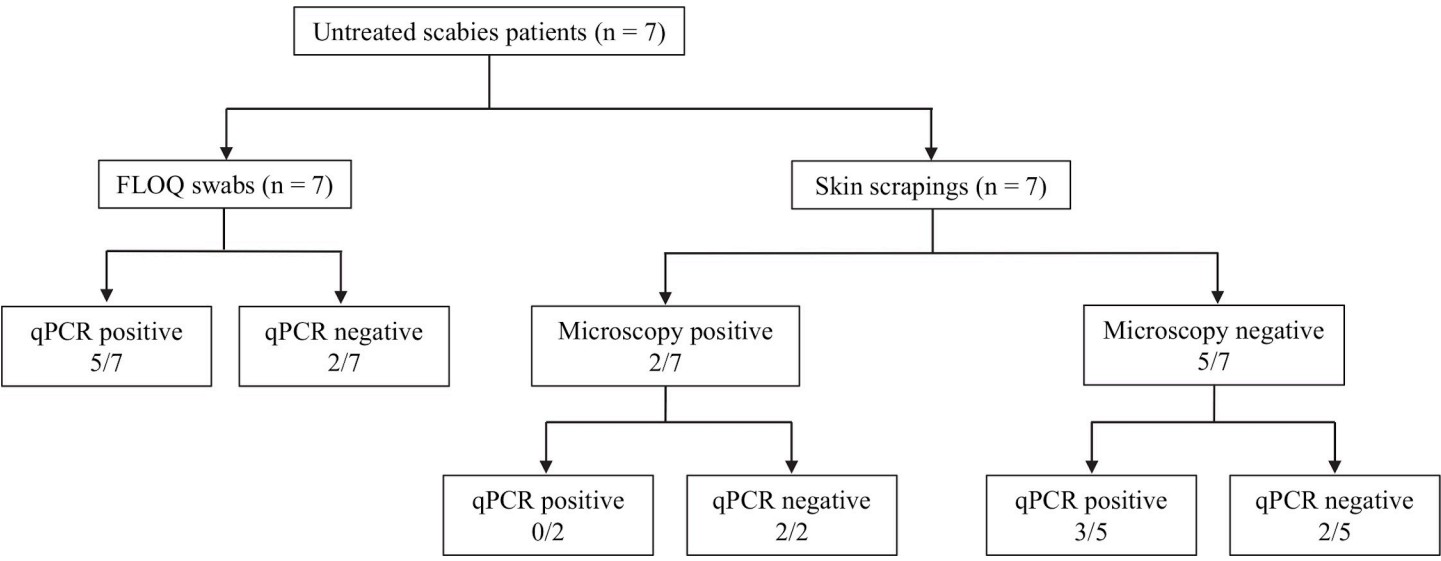

**Fig 5. Comparison of qPCR results on samples collected by FLOQ swabs and skin scrapings from 7 untreated scabies patients.**

SSR6 of 0.3 ± 1.62). Of note, investigation of qPCR assay performance in terms of cycle threshold suggested that sampling by FLOQ swab (7 positive) was more sensitive than skin scrapings, with five of seven FLOQ swabs showing a lower Cq than skin scrapings.

## Samples from school children in remote Aboriginal communities

A total of 15 school children were enrolled for scabies sampling in community surveys in Kimberley, Western Australia from September to November 2019 (S6 Table). All 15 patients (100%) were of Indigenous ethnicity. The median age was 7.5 years (interquartile range, IQR:

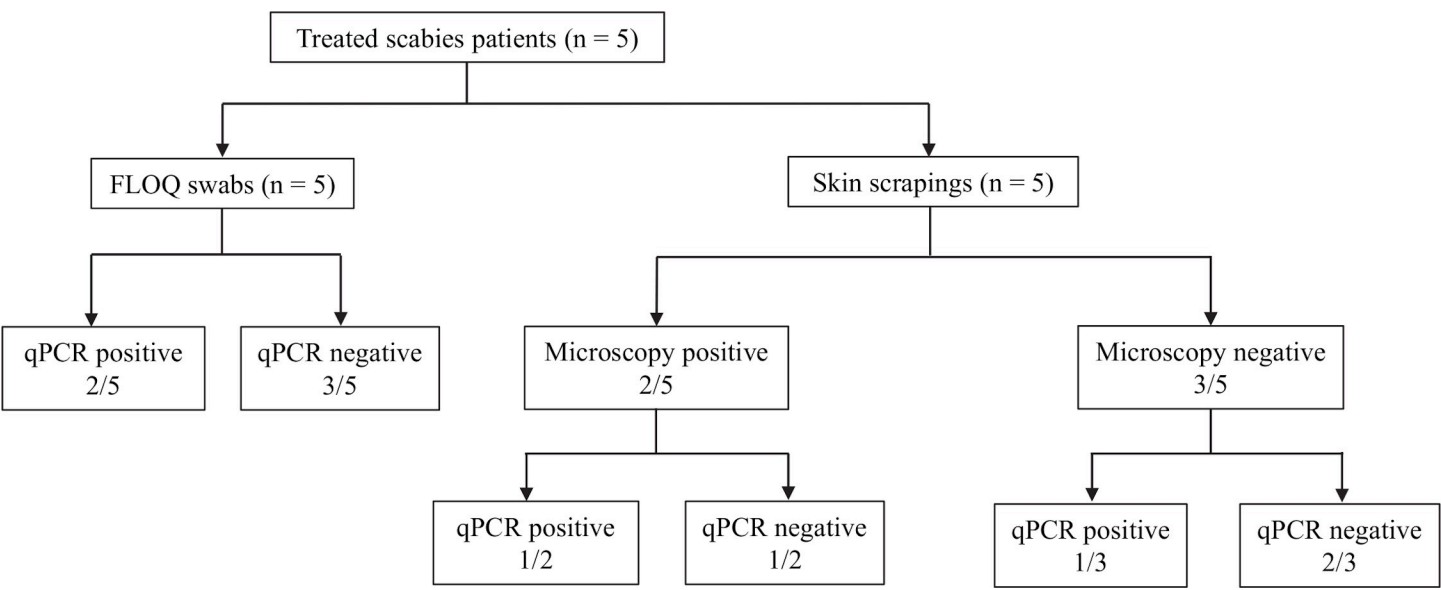

**Fig 6. Comparison of qPCR results on samples collected by FLOQ swabs and skin scrapings from 5 treated patients.**

**Table 2. Comparison of assay performance among patients with a positive qPCR result.**

| Patient no. | Sample Type | SSR5 | SSR6 | cox1 |
| --- | --- | --- | --- | --- |
| | | Cq Mean | Cq Mean | Cq Mean |
| 3 | Skin Scraping | 22.04 | 23.15 | 22.56 |
| | FLOQ | 34.55 | NA* | 34.52 |
| 5 | Skin Scraping | NA | NA | NA |
| | FLOQ | 31.33 | 32.64 | 31.54 |
| 7 | Skin Scraping | 30.12 | 31.20 | 30.0 |
| | FLOQ | 28.16 | 29.58 | 29.0 |
| 8 | Skin Scraping | 29.16 | 30.86 | 30.18 |
| | FLOQ | 23.85 | 25.20 | 24.95 |
| 10 | Skin Scraping | 22.49 | 24.27 | 24.93 |
| | FLOQ | 20.51 | 22.38 | 22.03 |
| 11 | Skin Scraping | 20.66 | 22.16 | 22.62 |
| | FLOQ | 25.26 | 27.39 | 26.73 |
| 12 | Skin Scraping | NA | NA | NA |
| | FLOQ | 26.47 | 28.09 | 27.84 |

* **NA:** No amplification detected

6–9.5). All 15 children had a clinical diagnosis of suspected scabies according to the IACS criteria [29]; swabs were collected exclusively from body sites displaying visible lesions. A second FLOQ swab sample was taken from one child due to the presence of scabies lesions on an additional site. None of the children were known to have received acaricidal treatment prior to sampling. Of the 15 swab samples tested, one tested positive with the SSR5 assay, and two tested positive with the *cox1* assay (S7 Table). In neither case did the samples overlap.

## Discussion

This study describes the development of two probe-based diagnostic real time qPCR assays for scabies detection, using two newly identified repetitive DNA sequences as assay targets. Application of a specific primer-probe pairing for the detection of abundant DNA targets showed enhanced sensitivity compared to routine microscopy in clinical samples. For the first time, we report the use of the FLOQ swab methodology for safe, non-invasive skin sampling with results from two high -burden scabies settings namely- an urban clinic and remote school-based surveillance. We have also demonstrated that these qPCR assays were more sensitive in detecting *S. scabiei* on samples collected by FLOQ swabs than on skin scrapings on clinically diagnosed scabies cases.

The results presented in this study concur with findings from previous studies [13,24], where PCR-based assays were reported to outperform traditional microscopic diagnosis. In this study, the SSR5 assay and the SSR6 assay demonstrated consistent detection of serially diluted mite genome DNA up to 1:100 of a single mite, presenting evidence of assay potential to detect minimal mite material from patient samples. Additionally, both SSR5 and SSR6 assays displayed comparable performance to a qPCR assay targeting the mitochondrial target *cox1* [13], which was selected as a reference target due to its reported high sensitivity. In particular, one of the two assays (SSR5) appears to be more sensitive than the *cox*1 qPC*R* assay, displaying lower Cq values throughout. qPCR diagnosis on clinical samples collected in in a clinical setting displayed potential as a diagnostic test, complementing microscopy and clinical diagnosis.

To enable non-invasive sample collection methods, we also investigated the suitability of swab sampling as an alternative to skin scrapings. In previous studies conducted by Fraser *et al.* (2018b), Catch-All swab sampling was associated with an increase in false-negatives and insufficient uptake of mite material for amplification. It was then suggested that cyto-brushes could eliminate aforementioned problems. Here, we have demonstrated this with the encouraging application of FLOQ swabs in the collection of samples in a clinical setting. In concordance with several studies utilising similar swab types [26,40,41], enhanced performance of the FLOQ swabs was evident when applied for sampling. Incremental benefit from swab collections could have a greater impact on PCR-based diagnostic sensitivity, promoting its capability as a suitable alternative to skin scrapings. Importantly, the combination of swab collections and PCR-based diagnostics do not rely on the collection of whole mites, which is necessary in the microscopic diagnosis of skin scrapings [42].

The potential application of sample collection at a remote site was investigated through the collection of FLOQ swab samples from suspected scabies cases in remote communities of Australia. It was observed that detection rate of the assays was lower on swab samples collected from school children with suspected scabies in a more remote epidemiologic setting in Kimberley compared to a better resourced clinical setting. Although a clinical diagnosis of scabies was made by trained research staff, no confirmatory test was performed on these children. Despite adherence to the IACS diagnostic criteria (S1 Table), misclassifications are possible, thus underlining the critical importance of undertaking reference diagnostic tests, such as dermatoscopy or microscopy on skin scrapings in studies evaluating performance of diagnostic tests, even if undertaken in remote settings. In further work to better define diagnostic sensitivity and specificity of molecular diagnostics for scabies it will be essential to include such gold standard methods.

Additionally, the lack of standardised sampling methods could greatly influence diagnostic performance, which was evident in a recent study [25]. Ideally, sampling should be carried out on distinctive lesions induced via parasite entry to obtain mite material. Identification of such lesions is dependent on training and skill of the clinician [3] or with the aid of a dermatoscope. A more generalised inflammatory eruption of papules can occur on skin separate from the locations of mite activity [43], and failure to obtain samples from lesions associated with infection will therefore result in a false negative test. Therefore, to better define assay sensitivity, it will be important to test the assay in clinical groups where diagnosis has been confirmed by dermatoscopy and/or microscopically. The promising performance of the FLOQ swab method in this small study suggests that it would be useful to confirm these findings in a larger study.

Scabies patients often continue showing symptoms after treatment, even in the absence of viable mite material, due to the hypersensitive immune response [44]. A confirmatory test could be useful in evaluating the success of scabies treatment, particularly in treatment studies. Amongst the treatment group in this study, most patients with prior treatment tested negative, with the exception of two patients, one in relatively early stages of treatment and another suffering from crusted scabies. Both these patients may have harboured a high parasite load. Evaluation of the kinetics of post-treatment clearance of mite DNA by qPCR with further validation studies could assist in understanding the role of qPCR in treated patients.

The identification of the abundant non-coding repetitive DNA sequences in the *S. scabiei* genome represents an avenue for further research on the genetics and molecular epidemiology of scabies [45]. When similar PCR diagnostic assays were developed for helminths, challenges in their application targeting repetitive elements as assay targets ensued in the Ascaridae species due to a process known as chromosome diminution [21], where repetitive DNA elements are eliminated from the genome during development [46,47]. Scabies patient samples could contain variations of mite components, including different mite life stages or faecal material.

Presently, it is unclear how these repetitive DNA sequences vary across the mite life cycle, and whether this would affect sensitivity of the PCR diagnosis assays. Future research directed towards identifying the function and development of repetitive DNA elements in the *S. scabiei* genome should aid in building confidence towards their application as diagnostic targets.

As scabies control efforts continue to expand, scarcity of diagnostic options hinders planning and implementation of elimination efforts. Ongoing development of reproducible and reliable methods remain a priority. Studies are required for further validation of these assays with additional clinical samples and sequential monitoring of recruited patients, to evaluate the diagnostic accuracy of qPCR in scabies, both pre- and post-treatment. To improve the probability of detection, the utility of taking multiple swabs per individual may also need to be considered. Future establishment of a definitive laboratory diagnostic test, along with the application of efficacious sampling methods, will thus hold the potential to improve clinical diagnosis of scabies in both remote and clinical settings, thereby contributing to improved disease surveillance. Accuracy and efficacy in scabies diagnostics could therefore significantly contribute to timely treatment implementations, facilitating the success of control programmes.

## Supporting information

**S1 Fig. Probe-based qPCR assays detection limits and efficiency.** qPCR amplification curves of **(A)** SSR5 and **(B)** SSR6 generated from serial 10-fold linearised plasmid DNA template dilutions ranging from $3 \times 10^8$ to $3 \times 10^1$ copies/μL. Each dilution was tested in triplicate. Blue line across indicates positive threshold. Linear standard curves of **(C)** the SSR5 assay and **(D)** the SSR6 assay constructed with triplicate cycle quantification (Cq) values against log starting DNA template quantity of each dilution. The equation for the regression line and the coefficient of determination ($R^2$) value are shown in the graphs. Detection limit was found to be at 3.8 copies per 10 μL of reaction for the SSR5 assay and 1.9 copies per 10 μL of reaction for the SSR6 assay. *RFU: relative fluorescence unit.
(PDF)

**S2 Fig. SSR5 and SSR6 Primer-Specificity.** Amplification **(A)** SSR5 and **(B)** SSR6 with gDNA panel consisting of human skin, pig skin, *P. humanus*, *D. canis*, *P. ovis* and serially-diluted HDMs. Neat, 1:10, 1:100 dilutions replicate presence of 100 (HDM100), ten (HDM10) and one (HDM1) HDM respectively. Amplification of **(C)** SSR5 and **(D)** SSR6 with gDNA of common skin pathogens including *S. pyogenes*, *S. aureus*, *P. acnes*, *T. rubrum*, *T. interdigitale* and *Trichophyton sp*. Negative control = water; Positive control = neat mite gDNA (n = 5).
(PDF)

**S1 Table. Summary of the 2018 IACS criteria for the diagnosis of scabies.**
(PDF)

**S2 Table. Primer and probe details for qPCR assays.**
(PDF)

**S3 Table. RepeatExplorer2 identified repetitive DNA elements with at least one high confidence call amongst four Illumina generated mite sequences.**
(PDF)

**S4 Table. qPCR results for the two-fold dilution series used to determine the assays limit of detection.**
(PDF)

**S5 Table. Triplicate cycle quantification (Cq) values for mite extractions to determine assay sensitivity.**
(PDF)

**S6 Table. Profile of school children from StoP Trial in Kimberley, WA enrolled in this study.**
(PDF)

**S7 Table. Diagnostic results of StoP Trial samples by qPCR.**
(PDF)

## Acknowledgments

We thank all the patients from Royal Darwin Hospital and Darwin Dermatology who participated in the study, the Clinical team at RDH: Sudharsan Venkatesan, Angela Wilson and Anja Hohls for collection of patient samples and the Menzies School of Health Research team: Vanessa Rigas and Celeste Woerle for sample processing in Darwin; the Queensland Animal Science Precinct team for maintenance of the scabies pig models.

The SToP trial is a partnership between the Telethon Kids Institute, WA Country Health Services–Kimberley, Kimberley Aboriginal Medical Services and Nirrumbuk Aboriginal Environmental Health Service. We acknowledge study staff involved in data collection: Marianne Mullane, Rachael Donovan and Frieda McLoughlin. We acknowledge the children and families of the Kimberley who participated in this study.

We also thank Dr. Laura Cascales for providing medical writing services; Dr. Gunter Hartel and Dr. Louise Marquart for offering statistical advice from QIMR Berghofer.

## Author Contributions

**Conceptualization:** James S. McCarthy, Cielo Pasay.

**Formal analysis:** Lena Chng, James S. McCarthy, Cielo Pasay.

**Funding acquisition:** Bart J. Currie, James S. McCarthy, Cielo Pasay.

**Investigation:** Lena Chng, Deborah C. Holt, Matt Field, Joshua R. Francis, Dev Tilakaratne, Milou H. Dekkers, Greg Robinson, Rebecca Pavlos, Katja Fischer, Anthony T. Papenfuss, Robin B. Gasser, Pasi K. Korhonen, Bart J. Currie, Cielo Pasay.

**Methodology:** Lena Chng, Deborah C. Holt, Asha C. Bowen, Bart J. Currie, James S. McCarthy, Cielo Pasay.

**Project administration:** Cielo Pasay.

**Resources:** Milou H. Dekkers, Kate Mounsey, Katja Fischer, Anthony T. Papenfuss, Robin B. Gasser, Pasi K. Korhonen.

**Supervision:** Deborah C. Holt, Joshua R. Francis, Rebecca Pavlos, Asha C. Bowen, Bart J. Currie, James S. McCarthy, Cielo Pasay.

**Writing – original draft:** Lena Chng, James S. McCarthy, Cielo Pasay.

**Writing – review & editing:** Lena Chng, Deborah C. Holt, Matt Field, Joshua R. Francis, Dev Tilakaratne, Milou H. Dekkers, Greg Robinson, Kate Mounsey, Rebecca Pavlos, Asha C. Bowen, Katja Fischer, Anthony T. Papenfuss, Robin B. Gasser, Pasi K. Korhonen, Bart J. Currie, James S. McCarthy, Cielo Pasay.

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
