## [Decision Letter · Decision Letter 0]

23 Sep 2020

Dear Dr Pasay,

Thank you very much for submitting your manuscript "Molecular diagnosis of scabies using a novel, probe-based polymerase chain reaction assay targeting high-copy number repetitive sequences in the Sarcoptes scabiei genome" for consideration at PLOS Neglected Tropical Diseases. As with all papers reviewed by the journal, your manuscript was reviewed by members of the editorial board and by several independent reviewers. In light of the reviews (below this email), we would like to invite the resubmission of a significantly-revised version that takes into account the reviewers' comments. 

We cannot make any decision about publication until we have seen the revised manuscript and your response to the reviewers' comments. Your revised manuscript is also likely to be sent to reviewers for further evaluation.

Sincerely,

Richard Stewart Bradbury, PhD

Associate Editor

Michael Marks

Deputy Editor

Reviewer's Responses to Questions

**Key Review Criteria Required for Acceptance?**

**Methods**

-Are the objectives of the study clearly articulated with a clear testable hypothesis stated?

-Is the study design appropriate to address the stated objectives?

-Is the population clearly described and appropriate for the hypothesis being tested?

-Is the sample size sufficient to ensure adequate power to address the hypothesis being tested?

-Were correct statistical analysis used to support conclusions?

-Are there concerns about ethical or regulatory requirements being met?

Reviewer #1: (No Response)

Reviewer #2: (No Response)

Reviewer #3: The authors present promising findings but correctly indicate that a larger validation study needs to be performed to better estimate diagnostic sensitivity.

There are no apparent ethical concerns.

Reviewer #4: see below

**Results**

-Does the analysis presented match the analysis plan?

-Are the results clearly and completely presented?

-Are the figures (Tables, Images) of sufficient quality for clarity?

Reviewer #1: (No Response)

Reviewer #2: (No Response)

Reviewer #3: The results are clearly expressed. The image quality of Figs 1 and 3, however, need improving.

Reviewer #4: see below

**Conclusions**

-Are the conclusions supported by the data presented?

-Are the limitations of analysis clearly described?

-Do the authors discuss how these data can be helpful to advance our understanding of the topic under study?

-Is public health relevance addressed?

Reviewer #1: (No Response)

Reviewer #2: (No Response)

Reviewer #3: The conclusions are supported by the data and limitations of the study are well discussed.

The public health relevance is highlighted well.

Reviewer #4: see below

**Editorial and Data Presentation Modifications?**

Reviewer #1: None

Reviewer #2: (No Response)

Reviewer #3: As above, the image quality of Figs 1 and 3 needs improving.

Reviewer #4: see below

**Summary and General Comments**

Reviewer #1: The results of this study show an improvement in diagnostic laboratory detection of scabies using PCR over standard microscopic examination of skin scrapings. The work was thorough and presented clearly. However, after reading the paper, I was left wondering what the role of laboratory diagnosis is in this setting since even with very sensitive molecular techniques, there was a gap between positive lab tests and clinical suspicion based on standardized criteria. If there is clinical suspicion but a laboratory test, even a highly-sensitive molecular test, is negative, how would a clinician respond? Although an improvement, PCR may still not be a totally reliable laboratory test. I think that the background information needs to more clearly state what the role of laboratory testing is given the low sensitivity even with the new molecular test described.

Minor comments:

p. 9 Consider changing “carer” to “caregiver”

Define what “NA” means in Tables S3, S4, and S5

P. 22 Figure 4 legend: Correct “”…before sample were taken…” to “… before samples were taken…”. The same correction is needed in the legend for Table 2. Also in the legend for Table 2, “ifestation” needs to be corrected to “infestation”.

Reviewer #2: Chng et al. Molecular diagnosis of scabies using a novel, probe-based polymerase chain reaction assay targeting high-copy number repetitive sequences in the Sarcoptes scabiei genome.

This study reports on the use of a new PCR method and the use of a new type of swab (as an alternative to skin scrapings) for the diagnosis of S. scabiei infections. This is a good study and by a well renowned group working in the difficult area of scabies diagnosis and treatment. Diagnosis of scabies/mange is a persistent challenge for humans and many animal species, and advancements in diagnosis are therefore of significant value. It is notable that improvements in the successful collection of mite material from infected individuals is perhaps the most valuable advancement that can be made for S. scabiei diagnosis. While this study focusses primarily on diagnostic method improvement, it also contributes some advancement in improved collection of mite material.

General feedback

To me the demonstrated value of FLOQ swabs is arguably the most important contribution of this study, and could perhaps be emphasised more strongly (see above and my specific comments). 

The biggest challenges I encountered reading this manuscript was keeping track of the varied components/sections of the study, and also variation in reporting results across the components, which wasn’t always clear as to why. Some of my specific feedback below reflects these challenges. 

Given the main limitation to diagnosing S. scabiei infection in patients is the acquisition of mite material (through skin scrapes or swabs), it would seem intuitive to always bring results back to the amount of mite material needed for a positive diagnosis. By this I mean represent all Cq and DNA copies results also as the amount of mite material required for diagnosis. You do this in Figure 2, but it should also be in Figure 3, Table 5 and through the results and discussion text (most notably p21, 27, 31).

The section of this study on Aboriginal school children is quite inconclusive and therefore a limitation of the study. It read to me that either: sample collection was insufficient for a fair PCR diagnosis; or more samples per individual are needed for a higher detection probability; or the criteria for a clinical diagnosis was not applied as well as needed. I do not know the answer to this, and I was left wondering how meaningful this section was. If it is so inconclusive, should it even be included in the study?

Specific feedback

There are quite a few components to this study, types of samples collected, and testing undertaken. I think the ms could be made more clear through the inclusion of a workflow diagram documenting these details. 

Methods: I note repetition of content in the Ethics Statement and Study Design sections. These paragraphs need revision to eliminate the repetition. 

Table 2: Change the “Diagnosis” column heading to “Clinical diagnosis”

P25. It is surprising that you found qPCR on FLOQ swabs to be more sensitive for detecting S. scabiei than qPCR on skin scrapings. This would seem contrary to previous literature, and I was left wondering if this result was genuine or more a reflection of the way the sentence presented the results. Assuming it is genuine, this is an important diagnostic method result. It would be valuable to present a table comparing positives and negatives between the methods, and associated sensitivity and specificity. 

Table 3: Why not report the pig model results in this table also? Afterall, you have 24 samples (spanning four individuals) from pigs, plus controls.

Discussion, paragraph 1: It is neither surprising nor novel that PCR was more sensitive than microscopy. For this reason I think a real value of this study is in reporting the use of FLOQ swabs.

Discussion, paragraph 3: Was qPCR done on skin scrapings to compare with swabs? This is a more meaningful comparison than microscopy for methods improvement. See also my comments regarding p25.

Discussion, paragraph 4: See my above comments about this being a major limitation of the study.

Discussion, final paragraph: The authors should also highlight that multiple swabs may need to be taken per individual to improve the detection probability for S. scabiei. This appears to be an underrepresented consideration (and valuable future direction) for use of FLOQ in S. scabiei diagnosis.

The authors might also note LAMP as an additional diagnostic method evaluated for S. scabiei: Fraser et al. PeerJ 6:e5291.

Reviewer #3: This paper presents novel preliminary findings of a newly developed qPCR assay, combined with a non-invasive sampling technique, to enhance the diagnosis of scabies. The public health importance of such research in scabies is very high. The authors acknowledge the limitations of the present study and plan a larger validation study.

Reviewer #4: This is an interesting paper outlining the development and assessment of 2 PCR assays for the diagnosis of scabies. The authors also highlight the superior performance of FLOQ swab samples when using PCR assays for scabies diagnosis.

There are several revisions that I could suggest to improve the manuscript:

1) abstract - include sensitivity in other clinical settings (treated patients / school children) for balance

2) Author summary -requires revision 

- "for progammatic use" - consider excluding

- ""Through new insights into.." - revise sentence - "We developed two PCR based diagnostic assays targeting..... based on new insignts..."

- "The use of such elements" - reword for clarity - maybe split into two sentences?

- deleted "established"

- revise "should provide a definitiive diagnosis in a clinical context" - perhaps "may provide a useful additional tool in the clinical diagnosis of scabies"

3) Introduction

- re-word "diagnostic deficiencies" - is this in reference to "the limitations of available methods of diagnosis"

- define abbreviation "qPCR"

- review "However, only the assay targeting the mitochondrial..." - perhaps "To date, only the assay..."

4) methods

- clarify who performed microscopy on skin scraping samples

5) results

- re "Comparative testing of FLOQ and Catch all swabs" - presumablt the values for DNA copies are mean values? (7351.1 and 3782,1 etc) - clarify in text

- please present overall sensitivity in Darwin context (treated and non-treated combined)

- can you calculate a sensitivity for the Kimberley school children's context? compared with clinical diagnosis - later discuss limitations of gold standard as suggested

- the sentence "Lower sensitivity of the qPCR assays..." is incorrect with regard to treated patients (sens 40% for skin scrapings was the same as for swab) - revise sentence

6) Tables

- table 3- present overall numbers (treated and non-treated combined) - consider including PPV and NPV (as outlined in methods?)

- table 4 - some overlap with table 3 - consider combining - alternatively exclude columns for "no scabies" - but rather report total - consider providing percentage positive in brackets next to numbers for microscopy and qPCR

- table 6 - doesn't add anything over and above text - consider excluding

- tables 7 - again doesn't add anything over and above text - I would exclude.

7) discussion

- re the sentence "This was ovserved in clinical patients from Darwin who were diagnosed with an early infection of scabies" - was this included in the results - if not avoid introducing new data in the discussion

- discuss with more clarity the possibility of "misclassification" of scabies diagnosis in the the context of the schools study - perhaps with a reference on the limitations of clinical diagnosis?

- revise the sentence "A confirmatory test would be very useful to evaluate the success of scabies treatment" - given the relatively low sensitivity and the possibility of persistent DNA in treated cases I'm not sure the results support this statement

- include more complete discussion of the limitations of this study including: small numbers, heterogenous population assessed (treated / non-treated), lack / limitations of "gold standard" diagnostic method / reliance on clinical diagnosis, etc

- in concluding remarks (final paragraph) - and in the discussion generally - re-iterate the implications of the findings of this study in the context of scabies diagnosis - my take from reading is that this may have an additional role in improving accuracy of clinical diagnosis of scabies? possibly have a role in follow up of treatment? - not sure of role in large scale diagnostic studies but this warrants further investigation..

PLOS authors have the option to publish the peer review history of their article (what does this mean?). If published, this will include your full peer review and any attached files.

Reviewer #1: No

Reviewer #2: Yes: Scott Carver

Reviewer #3: No

Reviewer #4: No
---

## [Decision Letter · Decision Letter 1]

7 Dec 2020

Dear Dr Pasay,

Thank you very much for submitting your manuscript "Molecular diagnosis of scabies using a novel, probe-based polymerase chain reaction assay targeting high-copy number repetitive sequences in the Sarcoptes scabiei genome" for consideration at PLOS Neglected Tropical Diseases. As with all papers reviewed by the journal, your manuscript was reviewed by members of the editorial board and by several independent reviewers. In light of the reviews (below this email), we would like to invite the resubmission of a significantly-revised version that takes into account the reviewers' comments. 

This version of manuscript is greatly improved on the first submission. I am concerned that the authors have not sufficiently addressed the comments of Reviewer #2 and request that they properly attend to these reasonable requests for amendments in any re-submission they make of the manuscript.

We cannot make any decision about publication until we have seen the revised manuscript and your response to the reviewers' comments. Your revised manuscript is also likely to be sent to reviewers for further evaluation.

Sincerely,

Richard Stewart Bradbury, PhD

Associate Editor

Michael Marks

Deputy Editor

This version of manuscript is greatly improved on the first submission. I am concerned that the authors have not sufficiently addressed the comments of Reviewer #2 and request that they properly attend to these reasonable requests for amendments in any re-submission they make of the manuscript.

Reviewer's Responses to Questions

**Key Review Criteria Required for Acceptance?**

**Methods**

-Are the objectives of the study clearly articulated with a clear testable hypothesis stated?

-Is the study design appropriate to address the stated objectives?

-Is the population clearly described and appropriate for the hypothesis being tested?

-Is the sample size sufficient to ensure adequate power to address the hypothesis being tested?

-Were correct statistical analysis used to support conclusions?

-Are there concerns about ethical or regulatory requirements being met?

Reviewer #2: (No Response)

Reviewer #4: no concerns

**Results**

-Does the analysis presented match the analysis plan?

-Are the results clearly and completely presented?

-Are the figures (Tables, Images) of sufficient quality for clarity?

Reviewer #2: (No Response)

Reviewer #4: no concerns

**Conclusions**

-Are the conclusions supported by the data presented?

-Are the limitations of analysis clearly described?

-Do the authors discuss how these data can be helpful to advance our understanding of the topic under study?

-Is public health relevance addressed?

Reviewer #2: (No Response)

Reviewer #4: no concerns

**Editorial and Data Presentation Modifications?**

Reviewer #2: (No Response)

Reviewer #4: no concerns

**Summary and General Comments**

Reviewer #2: Thank you for the opportunity to review this manuscript. I think that it has potential to make valuable contributions to (1) the manner in which S. scabiei are sampled from patients and (2) the subsequent diagnostic testing of the samples.

I thank the authors for the manuscript revisions they have made. This has improved the study. I am a little disappointed that the authors showed resistance some of my recommended revisions. I was left with the impression that some justifications for not considering appeared more as excuse making, and some responses where changes were made were a bit minimal. Below are the recommendations I made on the original review (or follow-on comments) that I would like the authors to consider more carefully.

To me the demonstrated value of FLOQ swabs is arguably the most clinically impactful contribution of this study, and could be emphasised more strongly.

In my original review I recommended the authors consider using the relationship between Cq values and amount of mite material presented in figure 2 to convert other Cq values (fig 3, table 5, p21, 27 and 31) to amount of mite material. My reasoning is that this is ultimately the most practical piece of information to someone attempting to sample mites. The authors have responded with a justification that this is not possible without additional research. I am still not entirely clear on that justification, since Cq values they report throughout the manuscript are consistent with the range presented in figure 2 and ‘estimated amount of mite material’ would seem a reasonable metric to report. I would like the authors to consider this more closely. If they really feel strongly that they cannot do this, then at the very least a paragraph in the discussion is needed about it.

I am pleased the authors considered my suggestion for a work-flow diagram. They have made a good start on this, and it should be included as a full figure in the manuscript (possibly at the expense of another figure). The format of the figure could be made more condensed to take up less page space, and also more information could be added on sample sizes, reasoning behind steps, key results, etc.

There are a several comments I made that reflect the challenges I had with integrating the results across Tables 3-5, and the authors responses reinforced that for me. Below are those original comments and author responses which illustrate this point. I would like to encourage the authors to re-consider the presentation of these tables to assist the reader with their integration.

1. MY COMMENT: P25. It is surprising that you found qPCR on FLOQ swabs to be more sensitive for detecting S. scabiei than qPCR on skin scrapings. This would seem contrary to previous literature, and I was left wondering if this result was genuine or more a reflection of the way the sentence presented the results. Assuming it is genuine, this is an important diagnostic method result. It would be valuable to present a table comparing positives and negatives between the methods, and associated sensitivity and specificity. AUTHOR RESPONSE: We have presented qPCR results on clinical samples collected by FLOQ swabs in Table 3 with associated sensitivity and specificity. The qPCR results on clinical samples collected by skin scrapings on the other hand, are presented in Table 4. In addition, Table 5 also shows qPCR performance being more sensitive in detecting S. scabiei on samples collected by FLOQ swabs than skin scrapings when we compared our assays’ performance on clinically diagnosed scabies cases. We therefore believe that collectively these results show that this is a promising diagnostic method.

2. MY COMMENT: Table 3: Why not report the pig model results in this table also? Afterall, you have 24 samples (spanning four individuals) from pigs, plus controls. AUTHOR RESPONSE: For assay development, we tested the efficacy of FLOQ swabs as a sample collection method on the pig models. In Table 3, we present the qPCR results of testing the swabs on clinical human samples hence, we did not combine the results.

3. MY COMMENT: Discussion, paragraph 3: Was qPCR done on skin scrapings to compare with swabs? This is a more meaningful comparison than microscopy for methods improvement. See also my comments regarding p25. AUTHOR RESPONSE: qPCR was performed on samples collected both by swabs and skin scrapings. Please see results presented in Tables 3 and 4 with associated sensitivity and specificity. Data presented in Table 5 further supports this comparison. (qPCR assay performance was more sensitive in detecting S. scabiei on samples collected by FLOQ swabs than skin scrapings from seven clinically diagnosed scabies cases).

Reviewer #4: I thank the authors for addressing the suggested amendments - I am satisfied that they have addressed all the relevant comments.

PLOS authors have the option to publish the peer review history of their article (what does this mean?). If published, this will include your full peer review and any attached files.

Reviewer #2: No

Reviewer #4: No
---

## [Decision Letter · Decision Letter 2]

15 Jan 2021

Dear Dr Pasay,

We are pleased to inform you that your manuscript 'Molecular diagnosis of scabies using a novel, probe-based polymerase chain reaction assay targeting high-copy number repetitive sequences in the Sarcoptes scabiei genome' has been provisionally accepted for publication in PLOS Neglected Tropical Diseases.

Best regards,

Richard Stewart Bradbury, PhD

Associate Editor

Michael Marks

Deputy Editor

Reviewer's Responses to Questions

**Key Review Criteria Required for Acceptance?**

**Methods**

-Are the objectives of the study clearly articulated with a clear testable hypothesis stated?

-Is the study design appropriate to address the stated objectives?

-Is the population clearly described and appropriate for the hypothesis being tested?

-Is the sample size sufficient to ensure adequate power to address the hypothesis being tested?

-Were correct statistical analysis used to support conclusions?

-Are there concerns about ethical or regulatory requirements being met?

Reviewer #2: (No Response)

**Results**

-Does the analysis presented match the analysis plan?

-Are the results clearly and completely presented?

-Are the figures (Tables, Images) of sufficient quality for clarity?

Reviewer #2: (No Response)

**Conclusions**

-Are the conclusions supported by the data presented?

-Are the limitations of analysis clearly described?

-Do the authors discuss how these data can be helpful to advance our understanding of the topic under study?

-Is public health relevance addressed?

Reviewer #2: (No Response)

**Editorial and Data Presentation Modifications?**

Reviewer #2: (No Response)

**Summary and General Comments**

Reviewer #2: I thank the authors for their attention to my suggestions. I sincerely appreciate the effort they have put into revisions on this manuscript, which I think makes an excellent contribution to scabies diagnostic research.

PLOS authors have the option to publish the peer review history of their article (what does this mean?). If published, this will include your full peer review and any attached files.

Reviewer #2: **Yes: **Scott Carver

---

## [Editor Report · Acceptance letter]

13 Feb 2021

Dear Dr Pasay,

We are delighted to inform you that your manuscript, "Molecular diagnosis of scabies using a novel, probe-based polymerase chain reaction assay targeting high-copy number repetitive sequences in the Sarcoptes scabiei genome," has been formally accepted for publication in PLOS Neglected Tropical Diseases.

Best regards,

Shaden Kamhawi

co-Editor-in-Chief

Paul Brindley

co-Editor-in-Chief
